# Potential of Ramalin and Its Derivatives for the Treatment of Alzheimer’s Disease

**DOI:** 10.3390/molecules26216445

**Published:** 2021-10-26

**Authors:** Tai Kyoung Kim, Ju-Mi Hong, Kyung Hee Kim, Se Jong Han, Il-Chan Kim, Hyuncheol Oh, Joung Han Yim

**Affiliations:** 1Division of Polar Life Sciences, Korea Polar Research Institute, Incheon 21990, Korea; tkkim@kopri.re.kr (T.K.K.); wnal5555@kopri.re.kr (J.-M.H.); kh313@kopri.re.kr (K.H.K.); hansj@kopri.re.kr (S.J.H.); ickim@kopri.re.kr (I.-C.K.); 2Department of Chemistry, Hanseo University, Seosan 31962, Korea; 3College of Pharmacy, Wonkwang University, Iksan 54538, Korea; hoh@wcu.ac.kr

**Keywords:** Alzheimer’s disease, ramalin, derivatives, therapeutic potential, antioxidant, β-secratase, anti-inflammatory

## Abstract

The pathogenesis of Alzheimer’s disease (AD) is still unclear, and presently there is no cure for the disease that can be used for its treatment or to stop its progression. Here, we investigated the therapeutic potential of ramalin (isolated from the Antarctic lichen, *Ramalina terebrata*), which exhibits various physiological activities, in AD. Specifically, derivatives were synthesized based on the structure of ramalin, which has a strong antioxidant effect, BACE-1 inhibition activity, and anti-inflammatory effects. Therefore, ramalin and its derivatives exhibit activity against multiple targets associated with AD and can serve as potential therapeutic agents for the disease.

## 1. Introduction

In 2016, the number of patients with dementia worldwide was 23.8 million, and this number is increasing every year owing to population aging and growth. Thus, caregiving and support for patients with dementia have far-reaching implications for families, health systems, and society as a whole [1,2]. Dementia is known to be associated with a shortened life expectancy, and this makes it the fifth leading cause of death in developed countries [3]. Further, Alzheimer’s disease (AD), which accounts for more than half this number of deaths, is the most common cause of dementia. Patients diagnosed with AD show progressively worsening symptoms, including memory disorders, speech disorders, spatiotemporal disorders, and mental disorders, including delusions and hallucinations. Furthermore, the pathogenesis of AD is diverse and complex. Low acetylcholine (ACh) concentrations in the synaptic cleft, amyloid beta (Aβ) peptide aggregation and accumulation, tangles of microtubule-associated protein tau, and oxidative stress are considered to be the main causes of AD. It has also been observed that Aβ aggregation and accumulation in the brain can be the result of increased Aβ production, decreased Aβ protease activity, or altered Aβ transport across the blood–brain barrier (BBB) [4]. Additionally, β-site APP-cleaving enzyme 1 (BACE-1), which is associated with an increase in Aβ production, is considered a target for the prevention and treatment of AD [5]. It has also been observed that Aβ is produced via the sequential cleavage of amyloid precursor protein (APP) by BACE-1 and BACE-2, with BACE-1 being the key enzyme that initiates the process [6]. Experimental evidence based on studies involving human APP transgenic mice suggests that memory deficits can be prevented via BACE-1 gene deletion [7], and reportedly Aβ aggregates can act as pathogenic mediators, inducing oxidative stress, neuroinflammation, and BACE-1 expression, which further exacerbate AD [8]. In addition to Aβ accumulation via BACE-1, oxidative stress and inflammation can also enhance Aβ production. In a study which showed the existence of an association between Aβ formation and inflammation, plaques were observed in the brains of monkeys with chronic inflammation [9]. Therefore, reducing inflammation based on the use of anti-inflammatory drugs could significantly lower the prevalence of AD, given that inflammation enhances the production of reactive oxygen species (ROS), which cause oxidative damage to lipids, proteins, and nucleic acids [10]. Additionally, nuclear factor kappa-light-chain-enhancer of activated B cells (NF-κB), which is related to inflammation, is involved in the expression of BACE-1, which is regulated by various mechanisms [11]. Thus, BACE-1, inflammation, and ROS are all related to Aβ formation from the initiation of AD to the post-onset period.

From another perspective, ROS-induced oxidative stress has emerged as a factor in the development of AD, and reportedly is responsible for the early onset and exacerbation of AD [12]. Mitochondrial ROS, which are produced under normal conditions, are balanced by antioxidants and related enzymes. However, dysfunction in this regard owing to oxidative injury results in the generation of high levels of ROS as well as the release of apoptosis-inducing proteins [13,14,15]. Additionally, mitochondrial endoplasmic reticulum dysfunction owing to oxidative stress induces a cellular stress response known as the unfolded protein response (UPR) [12,16,17], which can protect cells from the accumulation of toxic proteins, but can lead to neurodegeneration when sustained [14,18].

Ramalin, isolated from the Antarctic lichen, *Ramalina terebrata*, exerts potent antioxidant and antibacterial effects [19,20]. In our previous study, we investigated the various physiological activities of ramalin with the aim of developing it as a therapeutic agent for oxidation-related diseases, a functional food for anti-aging, and a functional cosmetic agent for whitening and wrinkle improvement [21]. Further, owing to its strong antioxidative power, it has also been shown to exhibit anti-inflammatory [22], lipid-accumulation-reducing [23], and anti-hepatic fibrosis [24] activities. Its BACE-1 inhibitory effects as well as its antioxidant and anti-inflammatory effects have been confirmed, suggesting that it has potential as a therapeutic agent for AD treatment. In this study, we investigated its physiological activity based on the modification of its structure by synthesizing its derivatives (Figure 1), which were obtained by replacing the functional group (phenolic hydroxyl) on the phenyl ring with either a methyl group or a fluorine atom.

Owing to various setbacks, no cure for AD has been established thus far. Further, the pathogenesis of the disease is not fully understood, and there is no cure for the disease that can be used for its treatment or to stop its progression. Therefore, to address this problem, we investigated the potential of ramalin and its derivatives, which do not target only a specific mechanism of action but exhibit various physiological activities, such as antioxidant, anti-inflammatory, and BACE-1 inhibitory activities, as therapeutic agents for AD.

## 2. Results

### 2.1. Synthesis of Ramalin and Its Derivatives

Ramalin was successfully synthesized according to the method developed in our previous study [25]. The synthesis was conducted by coupling phenylhydrazine to 1-benzyl-*N*-Cbz-l-glutamic acid, the starting material. After lowering the temperature to −5 °C, triethylamine (TEA) and ethyl chloroformate (ECF) were sequentially added to the solution of the starting material dissolved in dichloromethane (DCM). Thereafter, the reaction mixture was stirred for approximately 2 h, and while maintaining the temperature, hydrazine HCl salt was added to complete the coupling reaction. This was followed by the deprotection of the benzyl and benzyloxycarbonyl (Cbz) groups using Pd/C and H_2_ gas, after which the crude product was recrystallized to obtain ramalin. To obtain ramalin derivatives, similar reaction conditions were employed (Figure 1).

Reportedly, when ramalin is in the form of a dry solid, it is stable at room temperature (i.e., 25 °C); however, in an aqueous solution, it is unstable and this is considered a disadvantage [26]. Through HPLC analysis, the change in purity of ramalin and its derivatives (10 mM in water) at room temperature was measured by UV absorbance at 254 nm. The purity of ramalin decreased from 98% to 83% after 14 h, but no change in purity was observed in the derivatives. Therefore, to improve its stability and BBB penetration capacity, its structure was modified by replacing the phenolic hydroxyl groups at position two with either a methyl group or a fluorine atom and by changing their positions. Thus, several derivatives were obtained. Specifically, RA-2Me, RA-3Me, and RA-4Me were synthesized by changing the position of the methyl group, which is an electron-donating group, and RA-25Me and RA-34Me, with two methyl groups on the phenyl ring, were synthesized by introducing an extra methyl group. To investigate the effect of fluorine atoms, which are electron-withdrawing as opposed to the methyl group, RA-2F, RA-4F, RA-24F, and RA-pentafluoride (PF) were also successfully synthesized. Additionally, given that ramalin and its derivatives are low-molecular-weight antioxidants, it was likely that they will penetrate the BBB [27]. AD-related therapeutic agents are required to penetrate the BBB and reach their target. Therefore, to estimate BBB penetration capacity, the lipophilicity, number of hydrogen bonds, polar surface area (PSA), molecular weight (MW), etc., of ramalin and its derivatives were examined (Table 1) [28]. Drugs acting on the central nervous system that were introduced between 1983 and 2002 had an average MW of approximately 310 Da, and a MW < 450 Da has been suggested as a requirement for such drugs [28]. Given that ramalin and its derivatives are water-soluble low-molecular-weight compounds and that their lipophilicities and hydrophilicities balance out, their influence on AlogP, which indicates lipophilicity, was not significant. Additionally, given their PSA values (90 Å^2^ < PSA < 140 Å^2^), they are considered to be advantageous in terms of BBB penetration [28,29].

### 2.2. Antioxidant Effects of Ramalin and Its Derivatives

The in vitro antioxidant effects of ramalin (free radical scavenging ability, Fe^3+^ reducing power, superoxide anion scavenging ability, and tyrosinase inhibitory effects) have been confirmed in a previous study [20]. A 2,2-diphenyl-1-picryl-hydrazyl-hydrate (DPPH) assay was performed to verify the antioxidant effects of the newly synthesized ramalin derivatives (Table 1). Based on the results obtained, ramalin derivatives showed a higher antioxidant efficiency than commercially available butylated hydroxyanisole (BHA). Among the derivatives tested, the antioxidant effect of RA-2Me was found to be the most similar to that of ramalin. Further, the antioxidant effect of RA-PF showed a relatively high IC_50_ value. However, additional experiments are needed to determine whether the antioxidant activity decreases in the absence of a phenyl proton.

### 2.3. BACE-1 Inhibitory Activity of Ramalin and Its Derivatives

Slight BACE-1 expression can lead to the accumulation of Aβ, which causes AD, and this can be fatal given that AD is a chronic disease. Therefore, BACE-1 inhibition is a promising target for the suppression of AD. In this study, we used PanVera^®^’s BACE-1 fluorescence resonance energy transfer (FRET) assay kit (P2985, Madison, WI, USA) to confirm the BACE-1 inhibitory activity of ramalin and its derivatives [30]. The corresponding IC_50_ values of ramalin and its synthesized derivatives are shown in Table 2. The standard positive control agent used was commercially available LY2811376. The experiments were each performed in triplicates for all of the compounds. Thus, it was observed that except for RA-24F, ramalin and all its other derivatives showed IC_50_ values at micromolar concentrations. Specifically, the BACE-1 inhibitory activity of ramalin was 17.66 ± 2.74 µM, and among the derivatives RA-25Me showed the highest activity at 9.81 ± 1.21 µM.

### 2.4. Anti-Inflammatory Activity of Ramalin and Its Derivatives

The importance of inflammation in the pathogenesis of AD has been recognized. It has also been confirmed that inflammation plays a role in AD pathology and exacerbates the disease [31]. Thus, we investigated the cytotoxicity of ramalin and its derivatives and measured their nitric oxide concentration to confirm their anti-inflammatory activity. Ramalin was found to be slightly cytotoxic at the maximum concentration (Figure 2). Conversely, its derivatives exerted no cytotoxic effects on these cells. Further, ramalin showed nitrogen oxide (NO) inhibitory activity in a concentration-dependent manner. Its derivatives, including RA-25Me, RA-34Me, RA-2F, RA-4F, RA-24F, and RA-PF, also showed concentration-dependent NO inhibition, whereas RA-2Me, RA-3Me, and RA-4Me exerted weak anti-inflammatory effects. Additionally, among these derivatives, RA-24F showed a stronger anti-inflammatory effect than ramalin, without cytotoxicity.

## 3. Discussion

In this study, the antioxidant, BACE-1 inhibitory, and anti-inflammatory activities of ramalin and its derivatives were evaluated. Nine ramalin derivatives were synthesized with a relatively high yield (62–79%) under conditions similar to those that were employed for ramalin synthesis. The synthesized derivatives, possessing methyl groups or a fluorine atom, exerted antioxidant effects similar to or better than that of ramalin. Further, the antioxidant effect of the derivatives differed depending on the position of the methyl group, and derivatives with F atoms showed relatively low antioxidant effects. Characteristically, RA-2Me showed a higher antioxidant effect than ramalin. Further, most of the synthesized derivatives, except for RA-2Me, RA-3Me, and RA-4Me, inhibited NO in a concentration-dependent manner, without cytotoxicity, and RA-24F in particular showed a higher anti-inflammatory effect than ramalin, without cytotoxicity. Additionally, except for RA-24F, ramalin and its derivatives exhibited BACE-1 inhibitory activity at micromolar concentrations. Among them, RA-25Me showed the highest BACE-1 inhibitory activity at 9.81 ± 1.21 µM concentration.

No clear electronic effects can be observed for aromatic systems consisting of electron-withdrawing or -donating groups. However, when RA-2Me and RA-2F were compared, RA-2Me showed a superior antioxidant effect, while RA-2F showed good anti-inflammatory effects. Further, in terms of BACE-1 inhibitory activity, both the substances showed comparable results. Among the synthesized derivatives, RA-25Me and RA-34Me showed antioxidant, anti-inflammatory, and BACE-1 inhibitory activities similar to those of ramalin, but were better than ramalin in terms of cytotoxicity and stability. Conversely, RA-24F showed high anti-inflammatory activity; however, its BACE-1 inhibitory activity could not be confirmed.

Intensive efforts have been made to develop small-molecule BACE-1 inhibitors with sufficient inhibitory activity that can penetrate the BBB to reach the brain [32]. Even though therapeutic agents that target BACE-1 inhibition, which are big pharma favorites, can reduce Aβ levels, no improvement in cognitive function in AD patients has been observed [33], implying that early AD diagnosis and treatment are important. In recent studies, the development of small molecules with a multi-target approach centered on BACE-1, which considers the complexity of AD, has been reported [34]. Examples of such molecules include coumarin derivatives [35], which are inhibitors of BACE-1 and acetylcholinesterase (AchE), and curcumin derivatives [36,37], which are inhibitors of BACE-1 and glycogen synthase kinase 3 beta (GSK-3β). Therefore, compounds with multi-target activity are in the spotlight given that drugs that show only BACE-1 inhibitory activity present limitations. Further, memoquin, which exhibits inhibitory activities against BACE-1 and AChE as well as antioxidant effects, is a suitable example of a multi-target compound [38,39,40]. Spatial and episodic memory deficits in mice with scopolamine-induced amnesia have been resolved following memoquin treatment. Even though further confirmation is needed in this regard, antioxidant effects are necessary for substantial cognitive improvement as well as the clearance and suppression of Aβ production in AD. Prior to the onset of AD through Aβ accumulation, ROS-induced oxidative damage accelerates AD induction. Further, ROS cause oxidative damage to the nucleic acids, lipids, and proteins that are associated with AD induction, and these damaged substances have been identified in the brains of patients with AD in the early stages [41]. Reportedly, oxidative damage causes neuroinflammation and mediates BACE-1 activity [42,43], and Aβ produced by BACE-1 can, in turn, lead to ROS production, which further exacerbates the symptoms of AD [8,44,45]. Therefore, based on its various physiological activities, ramalin, a strong antioxidant, has considerable potential as a multi-target therapeutic agent for AD. However, further studies on its additional AD-inducer-related activities as well as those of its derivatives are needed. Therefore, the synthesis of derivatives with improved activity should be continued based on the results here presented.

## 4. Materials and Methods

### 4.1. General Experimental Information

All the solvent and reagents were obtained from the commercial suppliers Merck (Darmstadt, Germany) or TCI (Tokyo, Japan), and used without further purification. All glassware was thoroughly dried in a drying oven (60 °C) or flame and cooled down under a steam of dry argon just prior to use. A filter was obtained from the commercial supplier GE healthcare (GF/F, 0.7 µm, Whatman, UK). All reactions were carried out under an inert atmosphere of argon. Solvents and liquid reagents were transferred using a syringe. Organic extracts were dried over a drying agent, Na_2_SO_4_ and concentrated under reduced pressure with the aid of a rotary evaporator (Eyela, Tokyo, Japan). Residual solvents were removed under a high vacuum (Vacuubrand RZ 2.5, Wertheim, Germany, 1 × 10^−2^ mbar). Accurate mass spectra were obtained with an AB Sciex Triple TOF 4600 (Framingham, MA, USA) instrument with the interface in direct injection mode. The IR spectra were measured using a Nicolet 6700 (Thermo Nicolet, Madison, WI, USA). NMR data were collected on a Jeol JNM ECP-400 spectrometer (Jeol Ltd., Tokyo, Japan) with a mixture of D_2_O (with 0.01 mg/mL of DSS)-acetone-*d*_6_ (6:1 *v*/*v*) or DMSO-*d*_6_ (dimethyl sulfoxide-*d*_6_) as solvents and the internal reference or the residual solvent signals for referencing (D_2_O (with DSS)-acetone-*d*_6_: *d*H 0.00/*d*C 29.8, DMSO-*d*_6_: *d*H 2.50/*d*C 39.5). Peak splitting patterns were abbreviated as m (multiplet), s (singlet), d (doublet), t (triplet), dd (doublet of doublets), and td (triplet of doublets), respectively. A microplate reader (Thermo Scientific Inc., San Diego, CA, USA) and multimode plate reader (Multistkan^TM^ GO, Thermo Scientific, Waltham, MA, USA) were used for absorbance analysis.

### 4.2. Synthesis and Characterization

#### 4.2.1. General Method for the Synthesis of p-Glu-Hyd Analogues

A 250 mL round-bottom flask equipped with a magnetic stir bar was charged with 1-Benzyl-*N*-Cbz-l-glutamic acid (2.0 g, 5.39 mmol) in a DCM (50 mL) solvent. The reaction mixture was cooled to −5 °C and TEA (1.2 eq, 6.47 mmol, 902 µL) was added slowly. After 10 min, ECF (1.2 eq, 6.47 mmol, 615 µL) was added dropwise to the mixture over 1h. The reaction mixture was stirred for 4 h at −5 °C. Another 100 mL pear flask was prepared and charged with phenyl hydrazine HCl salt (1.2 eq, 6.47 mmol) and TEA (1.2 eq, 6.47 mmol, 902 µL), and then slowly added to the main reaction flask for 1 h at −5 °C. When the hydrazine addition was finished, the reaction mixture was warmed to room temperature and stirred for 16 h. After completion of the reaction, the organic layer was washed in the order of distilled water, 1 N HCl, 0.5 N NaHCO_3_, and distilled water to separate the layers, and the organic layer was collected. The organic phase was dried over Na_2_SO_4_ and concentrated in a rotary evaporator. Purification was achieved by recrystallization from ethyl acetate/n-hexane (1:5).

#### 4.2.2. General Method for the Synthesis of Ramalin Derivatives

A 500 mL round-bottom flask equipped with a magnetic stir bar was charged with an appropriate p-Glu-Hyd analogue (4.5 mmol) and palladium on carbon (10 wt%) in MeOH 200 mL. The mixture was stirred under a hydrogen atmosphere (1 atm, hydrogen balloon) for 16 h. Upon completion, the reaction mixture was filtered through glass microfiber filter paper (0.7 µm). The filtrate was concentrated in a rotary evaporator. Purification was achieved by recrystallization from MeOH/ethyl acetate (1:5). Detailed information are presented in the Appendix A.

N^5^-(o-Tolylamino)-l-glutamine (**RA-2Me**). From Benzyl-*N*-Cbz-l-glutamic acid; 0.94 g, 70%, white solid; IR (ν cm^−1^, KBr): 3273, 3030, 1653, 1584, 1521. ^1^H NMR (400 MHz, D_2_O/Acetone-*d*_6_ (6/1)): δ 7.17 (m, 2H, PhH), 6.90 (td, *J* = 7.6, 1.2, 1H, PhH), 6.84 (dd, *J* = 8.0, 0.8, 1H, PhH), 3.81 (t, *J* = 6.0, 1H, H-2), 2.56 (m, 2H, H-4), 2.22 (s, 3H, CH_3_), 2.21 (m, 2H, H-3); ^13^C NMR (100 MHz D_2_O/Acetone-*d*_6_ (6/1)): δ 174.8, 174.0, 145.2, 131.0, 127.4, 123.9, 121.3, 112.2, 54.5, 29.9, 26.5, 16.5; HRESIMS *m*/*z* 252.1352 [M + H]^+^ (calculated for C_12_H_18_N_3_O_3_, 252.1348).

N^5^-(m-Tolylamino)-l-glutamine (**RA-3Me**). From Benzyl-*N*-Cbz-l-glutamic acid; 0.97 g, 72%, white solid, IR (ν cm^−1^, KBr): 3265, 3037, 1649, 1609, 1593, 1513. ^1^H NMR (400 MHz, D_2_O/Acetone-*d*_6_ (6/1)): δ 7.19 (t, *J* = 7.6, 1H, PhH), 6.79 (d, *J* = 7.6, 1H, PhH), 6.72 (s, 1H, PhH), 6.69 (d, *J* = 8.0, 1H, PhH), 3.80 (t, *J* = 6.4, 1H, H-2), 2.54 (m, 2H, H-4), 2.27 (s, 3H, CH_3_), 2.20 (m, 2H, H-3); ^13^C NMR (100 MHz D_2_O/Acetone-*d*_6_ (6/1)): δ 175.0, 174.1, 147.7, 140.3, 129.8, 122.3, 114.2, 110.9, 54.5, 29.9, 26.5, 20.9; HRESIMS *m*/*z* 252.1367 [M + H]^+^ (calculated for C_12_H_18_N_3_O_3_, 252.1348).

N^5^-(p-Tolylamino)-l-glutamine (**RA-4Me**). From Benzyl-*N*-Cbz-l-glutamic acid; 0.90 g, 67%, white solid, IR (ν cm^−1^, KBr): 3238, 3025, 1651, 1614, 1512. ^1^H NMR (400 MHz, D_2_O/Acetone-*d*_6_ (6/1)): δ 7.15 (d, *J* = 8.4, 2H, PhH), 6.82 (d, *J* = 8.0, 2H, PhH), 3.79 (t, *J* = 6.4, 1H, H-2), 2.52 (m, 2H, H-4), 2.25 (s, 3H, CH_3_), 2.18 (m, 2H, H-3); ^13^C NMR (100 MHz D_2_O/Acetone-*d*_6_ (6/1)): δ 174.9, 174.1, 145.1, 131.5, 130.2, 114.1, 54.5, 29.9, 26.5, 19.9; HRESIMS *m/z* 252.1367 [M + H]^+^ (calculated for C_12_H_18_N_3_O_3_, 252.1348).

N^5^-((2,5-Dimethylphenyl)amino)-l-glutamine (**RA-****25Me**). From Benzyl-*N*-Cbz-l-glutamic acid; 1.12 g, 78%, white solid, IR (ν cm^−1^, KBr): 3257, 2929, 1624, 1509. ^1^H NMR (400 MHz, D_2_O/Acetone-*d*_6_ (6/1)): δ 6.89 (d, *J* = 7.6, 1H, PhH), 6.57 (d, *J* = 7.6, 1H, PhH), 6.52 (s, 1H, PhH), 3.67 (t, *J* = 6.4, 1H, H-2), 2.43 (m, 2H, H-4), 2.10 (s, 3H, CH_3_), 2.06 (m, 2H, H-3), 2.03 (s, 3H, CH_3_); ^13^C NMR (100 MHz D_2_O/Acetone-*d*_6_ (6/1)): δ 174.4, 173.7, 144.9, 137.1, 130.7, 121.5, 120.4, 112.5, 54.2, 29.6, 26.3, 20.4, 15.8; HRESIMS *m/z* 266.1507 [M + H]^+^ (calculated for C_13_H_20_N_3_O_3_, 266.1505).

N^5^-((3,4-Dimethylphenyl)amino)-L-glutamine (**RA-34Me**). From Benzyl-*N*-Cbz-l-glutamic acid; 1.01 g, 70%, white solid, IR (ν cm^−1^, KBr): 3256, 2919, 1652, 1615, 1506. ^1^H NMR (400 MHz, D_2_O/Acetone-*d*_6_ (6/1)): δ 7.06 (d, *J* = 8.0, 1H, PhH), 6.71 (d, *J* = 2.4, 1H, PhH), 6.64 (dd, *J* = 8.0, 2.4, 1H, PhH), 3.80 (t, *J* = 6.4, 1H, H-2), 2.53 (m, 2H, H-4), 2.20 (m, 2H, H-3), 2.19 (s, 3H, CH_3_). 2.15 (s, 3H, CH_3_); ^13^C NMR (100 MHz D_2_O/Acetone-*d*_6_ (6/1)): δ 174.7, 173.9, 145.7, 138.3, 130.6, 129.9, 115.3, 111.4, 54.5, 29.9, 26.6, 19.4, 18.3; HRESIMS *m/z* 266.1508 [M + H]^+^ (calculated for C_13_H_20_N_3_O_3_, 266.1505).

N^5^-((2-Fluorophenyl)amino)-L-glutamine (**RA-2F**). From Benzyl-*N*-Cbz-l-glutamic acid; 0.85 g, 62%, white solid, IR (ν cm^−1^, KBr): 3249, 3046, 1659, 1619, 1501. ^1^H NMR (400 MHz, D_2_O/Acetone-*d*_6_ (6/1)): δ 7.15–7.08 (m, 2H, PhH), 6.97–6.90 (m, 2H, PhH), 3.80 (t, *J* = 6.0, 1H, H-2), 2.55 (m, 2H, H-4), 2.20 (m, 2H, H-3); ^13^C NMR (100 MHz D_2_O/Acetone-*d*_6_ (6/1)): δ 175.0, 174.0, 151.5 (d, ^1^*J*_CF_ = 238.4, CF), 135.3 (d, ^2^*J*_CF_ = 10.9), 125.2 (d, ^4^*J*_CF_ = 3.5), 121.7 (d, ^3^*J*_CF_ = 6.9), 115.7 (d, ^2^*J*_CF_ = 17.9), 114.9 (d, ^3^*J*_CF_ = 2.5), 54.5, 29.9, 26.5; HRESIMS *m/z* 256.1059 [M + H]^+^ (calculated for C_11_H_15_FN_3_O_3_, 256.1097).

N^5^-((4-Fluorophenyl)amino)-L-glutamine (**RA-4F**). From Benzyl-*N*-Cbz-l-glutamic acid; 0.90 g, 65%, white solid, IR (ν cm^−1^, KBr): 3266, 3024, 1649, 1609, 1508. ^1^H NMR (400 MHz, D_2_O/Acetone-*d*_6_ (6/1)): δ 7.09–7.04 (m, 2H, PhH), 6.92–6.88 (m, 2H, PhH), 3.82 (td, *J* = 6.0, 1.2, 1H, H-2), 2.54 (m, 2H, H-4), 2.21 (m, 2H, H-3); ^13^C NMR (100 MHz D_2_O/Acetone-*d*_6_ (6/1)): δ 175.0, 174.1, 158.0 (d, ^1^*J*_CF_ = 234.8, CF), 143.7 (d, ^4^*J*_CF_ = 2.1), 116.1 (d, ^2^*J*_CF_ = 22.8), 115.2 (d, ^3^*J*_CF_ = 8.0), 54.4, 29.8, 26.4; HRESIMS *m/z* 256.1106 [M + H]^+^ (calculated for C_11_H_15_FN_3_O_3_, 256.1097).

N^5^-((2,4-Difluorophenyl)amino)-L-glutamine (**RA-24F**). From Benzyl-*N*-Cbz-l-glutamic acid; 1.10 g, 75%, white solid, IR (ν cm^−1^, KBr): 3271, 3043, 1655, 1609, 1508. ^1^H NMR (400 MHz, D_2_O/Acetone-*d*_6_ (6/1)): δ 6.99 (m, 1H, PhH), 6.94 (m, 1H, PhH), 6.89 (m, 1H, PhH), 3.79 (t, *J* = 6.4, 1H, H-2), 2.52 (m, 2H, H-4), 2.18 (m, 2H, H-3); ^13^C NMR (100 MHz D_2_O/Acetone-*d*_6_ (6/1)): δ 175.1, 174.1, 157.1 (dd, ^4^*J*_CF_ = 3.3, ^1^*J*_CF_ = 237.9, CF), 151.2 (dd, ^3^*J*_CF_ = 12.2, ^1^*J*_CF_ = 241.9, CF), 131.8 (dd, ^4^*J*_CF_ = 3.3, ^2^*J*_CF_ = 11.1), 115.8 (dd, ^3^*J*_CF_ = 3.9, ^3^*J*_CF_ = 9.4), 111.4 (dd, ^4^*J*_CF_ = 3.7, ^2^*J*_CF_ = 22.1), 104.3 (dd, ^2^*J*_CF_ = 22.3, ^2^*J*_CF_ = 26.9), 54.5, 29.8, 26.4; HRESIMS *m/z* 274.0963 [M + H]^+^ (calculated for C_11_H_14_F_2_N_3_O_3_, 274.1003).

N^5^-((Perfluorophenyl)amino)-l-glutamine (**RA-PF**). From Benzyl-*N*-Cbz-l-glutamic acid; 1.39 g, 79%, white solid, IR (ν cm^−1^, KBr): 3278, 3022, 1666, 1628, 1523. ^1^H NMR (400 MHz, DMSO-*d*_6_): δ 3.20 (t, *J* = 6.4, 1H, H-2), 2.27 (m, 2H, H-4), 1.85 (m, 2H, H-3); ^13^C NMR (100 MHz DMSO-*d*_6_): δ 172.0, 169.5, 137.3 (br d, ^1^*J*_CF_ = 246.0, 4C), 133.5 (br d, ^1^*J*_CF_ = 250.5, 1C), 124.8 (m, 1C), 53.5, 29.6, 26.9; HRESIMS *m/z* 328.0725 [M + H]^+^ (calculated for C_11_H_14_F_2_N_3_O_3_, 328.0721).

### 4.3. DPPH Assay (In Vitro)

According to the Blois, M.S. et al. method, the DPPH radical scavenging activity of ramalin and its derivatives was assessed [46]. Briefly, 150 µL of ramalin, its derivatives, and BHA at a 10, 5, 2.5, and 1 µM concentration solution in MeOH was mixed with 50 µL of 0.1 mM DPPH in MeOH, and then placed in the dark for 30 min at room temperature. Then, the mixture was determined at 540 nm.

### 4.4. Anti-Inflamation Activity Assay

#### 4.4.1. Cell Culture

The macrophage-like murine cell line RAW 264.7 (KCLB number 40071; Korean Cell Line Bank, Seoul, Korea) was cultured in Dulbecco’s Modified Eagle Medium (DMEM, Sigma-Aldrich, St. Louis, MO, USA) supplemented with 10% heat-inactivated fetal bovine serum (FBS, Invitrogen, Burlington, ON, Canada) and 1% (*w*/*v*) antibiotic–antimycotic solution (Invitrogen, Grand Island, NY, USA) at 5% CO_2_ and 37 °C.

#### 4.4.2. Cytotoxicity Assay

Cell cytotoxicity was determined by an MTT (3-(4,5)-dimethylthiazol-2-yl-2,5-diphenyltetrazolium bromide, Amresco, Solon, OH, USA) colorimetric assay. RAW 264.7 cells were seeded at a density of 2 × 105 cells/mL in 96-well plates and incubated in the presence of various concentrations of ramalin and its derivatives for 24 h. After incubation with test substances, MTT solutions (5 μL of a 5 mg/mL concentration in PBS) were added to the wells and incubated for 4 h at 37 °C. Then, treated with 100 μL of fresh DMSO to dissolve the crystals for 10 min, the cells were detected under a microplate reader (Thermo Scientific Inc., San Diego, CA, USA) to measure the absorbance at 570 nm. Relative cell viability was calculated by comparing the absorbance of the untreated control group. All experiments were performed in triplicate.

#### 4.4.3. Determination of Nitric Oxide Production

Nitrite accumulation was used as an indicator of NO production in the medium, and the nitrite level was determined by assaying the culture supernatants for nitrite using the Griess reagent (1% sulfanilamide, 0.1% *N*-(1-naphathyl)-ethylenediamine dihydrochloride, and 5% phosphoric acid). In order to measure the amount of nitrite, 1 × 106 cells/mL were seeded onto 96-well plates and then treated with the indicated concentrations of ramalin and its derivatives at 37 °C for 1 h, followed by stimulation with 0.5 μg/mL of lipopolysaccaride (LPS, 0.5 μg/mL, Sigma-Aldrich, CA, USA) for 24 h in a final volume of 200 μL. Then, 100 μL of cell culture supernatants were mixed with 100 μL of Griess reagent in a 96-well plate. Sodium nitrite was used to generate a standard curve and the concentration of nitrite was then measured by the absorbance at 540 nm using a microplate reader. All determinations were performed in triplicate.

### 4.5. β-Secretase (BACE-1) Inhibition Assay

A BACE-1 inhibition assay was conducted using a β-Secretase FRET kit (BACE-1, Thermo Fisher Sientific, San Diego, CA, USA) according to manufacturer instructions. The assay was carried out according to the manufacturer’s protocol as described earlier. A stock of ramalin and its derivatives in deionized distilled water (DDW) was prepared (20 mM). The sample was diluted further in assay buffer (final concentrations of 50, 25, 12.5, 6.25, 3.12, 1.56, 0.78, 0.39, 0.2, and 0.1 µM in each well) and black 96-well microplates were mixed with 10 µL of BACE-1 substrate. Then, 10 µL of 3 × BACE-1 enzyme was added to each well to start the reaction. Plates were incubated at room temperature in the dark for 60 min. After incubation, 10 µL of 2.5 mM sodium acetate was added to each well to stop the reaction. Finally, multiwell spectrofluorometer instruments were used for a multimode plate reader (Multistkan^TM^ GO, Thermo Scientific, Waltham, MA, USA) at an excitation wavelength of 545 nm and an emission wavelength of 585 nm. The half-maximal inhibitory concentration (IC_50_) was calculated by plotting the obtained relative fluorescence unit per hour (RFU/h) against the logarithmic of the inhibitor concentration. All determinations were performed in triplicate.

## 5. Conclusions

AD has a complex pathophysiology that includes pathological protein aggregation, neurotransmission disorders, increased oxidative stress, and microglia-mediated neuroinflammation. Therapeutic agents targeting only one of the molecular targets associated with AD have been reported. For example, therapeutic agents with high BACE-1 inhibitory activity can successfully suppress and eliminate Aβ production, but cannot restore cognitive function. Therefore, while developing therapeutic agents for AD, it is important to focus on multi-target drugs rather than single-target drugs. Ramalin, which possesses antioxidant, BACE-1 inhibitory, and anti-inflammatory activities, could serve as a promising therapeutic agent with multiple activities that could address the complex nature of AD. However, its application is limited owing to its low stability in aqueous environments and high-concentration cytotoxicity. RA-25Me and RA-34Me exhibited activity similar to that of ramalin, and highlighted the possibility of ameliorating these disadvantages. Therefore, the improvement of the physical properties and activities of ramalin via the synthesis of additional derivatives can lead to its application in AD treatment and prevention in future.

## Data Availability

The data presented in this study are available in this article.

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
