# Peer review of "Potential of Ramalin and Its Derivatives for the Treatment of Alzheimer’s Disease"

_molecules, 2021, doi:10.3390/molecules26216445_

Round 1

Reviewer 1 Report

The authors submitted an interesting manuscript dealing with the synthesis of ramalin derivatives and testing their inhibitory activity against BACE-1. In addition antioxidant effects and inhibitory effects on inflammation of ramalin and its derivatives were investigated. The aim of the study was to test the ability of ramalin and its derivatives to act on multiple targets associated with Alzheimer's disease. Given the seriousness and prevalence of the disease in the elderly population, the search for new effective drugs is highly desirable.

The synthesis of target compounds is well described. The structures were confirmed by 1H NMR, 13C NMR.

Methods for determining inhibitory, antioxidant and anti-inflammatory effects of ramalin and its derivatives are well described.

The results are supported by experimental data and they are discussed. Furthermore, the authors cited the recent articles published in this scientific field.

In my opinion, the submitted manuscript is complex. I recommend it for publication.

I have only 2 comments:

page 9, line 328: I assume there is a typographical error in the density notation. I think it should be 2×10<sup>5</sup>

page 9, line 340-341: the same problem, 3×10<sup>6</sup>

Author Response

Dear reviewer 1

Thank you for careful review and suggestions.

We would like to thank you and the reviewers for your insightful comments, which have greatly helped us to improve the quality of our manuscript. And we are honored to submit ‘Molecules’

Reviewer 1.

The authors submitted an interesting manuscript dealing with the synthesis of ramalin derivatives and testing their inhibitory activity against BACE-1. In addition antioxidant effects and inhibitory effects on inflammation of ramalin and its derivatives were investigated. The aim of the study was to test the ability of ramalin and its derivatives to act on multiple targets associated with Alzheimer's disease. Given the seriousness and prevalence of the disease in the elderly population, the search for new effective drugs is highly desirable.

The synthesis of target compounds is well described. The structures were confirmed by 1H NMR, 13C NMR.

Methods for determining inhibitory, antioxidant and anti-inflammatory effects of ramalin and its derivatives are well described.

The results are supported by experimental data and they are discussed. Furthermore, the authors cited the recent articles published in this scientific field.

In my opinion, the submitted manuscript is complex. I recommend it for publication.

I have only 2 comments:

page 9, line 328: I assume there is a typographical error in the density notation. I think it should be 2×10<sup>5</sup>

page 9, line 340-341: the same problem, 3×10<sup>6</sup>

  • We revised the paper by actively collecting the opinions of reviewers. Would you please review the revised content and advise if there are any problems?

We look forward to hearing from you.

Best regards,

Dr. Tai Kyoung Kim

Researcher, KOPRI

E-mail: tkkim@kopri.re.kr

Reviewer 2 Report

Reviewer have some comments on your manuscript as follows:

  1. The authors should check for English and typographical errors throughout the entire manuscript.
  2. Definition for all abbreviations in the text and that included in tables, figures should be given, such as: Aβ, Cbz, HBA, HBD, LPS, NO,…
  3. The abstract should be rewritten to be more condensed and convincing.
  4. In these statements “The pathogenesis of Alzheimer's disease (AD) is still unclear, and presently, there are no approved drugs that can be used for its treatment or to stop its progression” (lines 13-14), and “Further, the pathogenesis of the disease is not fully understood, and there are no approved drugs that can be used for its treatment or to stop its progression” (lines 86-88), the underline sentence is not true, and it could be rewritten as “there is no cure for the disease”.
  5. The statement “and can therefore serve as therapeutic agents for the disease” (line 25) should be rewritten as “and can therefore serve as potential therapeutic agents for the disease”.
  6. All tables and figures must be placed after their first citations in the text.
  7. In the Table 1, an additional column of molecular weight should be included.
  8. In the Table 2, the information of a control substance and its BACE-1 inhibitory activity should be included.
  9. In Figure 2: “% of contorl” should be corrected into “% of control”. The authors should specify what do the symbols #, **, and * denote for.
  10. Section 4.2.2: the authors should add the additional information on IR spectra, melting points, and solubilities of synthesized compounds.

For fluorine-containing derivatives (RA-2F, RA-4F, RA-24F, RA-PF), coupling constants: JH-F and JC-F should be added.

IR and MS spectra of synthesized compounds should be included in the Supplementary.

  1. Scheme 1: The information of the synthetic yields of synthesized compounds should be added.
  2. The authors should specify which synthesized compound(s) is (are) the new structure(s) with the appropriate citation (ie. Scifinder).
  3. Section 4.5. It should be indicated what the final concentrations of the tested compounds were in each well, what control compound was used and its concentrations in the wells.
  4. In the discussion, the authors stated that, except for the derivative RA-24F, ramalin and the rest of the derivatives inhibit BACE-1 in a concentration-dependent manner, but the reviewer found no evidence for this statement in the manuscript. Therefore, the authors should provide additional evidence to support this issue.
  5. The authors should also provide the evidence showing that ramalin derivatives are more stable than ramalin itself.
  6. In the discussion, the authors reported that the tendency to search for small molecules that act simultaneously on multiple therapeutic targets of AD is consistent with the multifactorial nature of AD. Currently, many studies have been conducted for different derivatives such as flavonoids, curcumin and on many targets such as AChE, BACE-1, antioxidant, GSK-3beta, etc. The authors could cite more recent works to support their arguments, such as:
  7. Tran T.-S., Le M.-T., Tran T.-D., Tran T.-H., Thai K.-M. (2020), “Design of Curcumin and Flavonoid Derivatives with Acetylcholinesterase and Beta-Secretase Inhibitory Activities Using in Silico Approaches”, Molecules, 25 (16), pp. 3644.
  8. Tran T.-S., Le M.-T., Nguyen T.-C.-V., et al. (2020), “Synthesis, In Silico and In Vitro Evaluation for Acetylcholinesterase and BACE-1 Inhibitory Activity of Some N-Substituted-4-Phenothiazine-Chalcones”, Molecules, 25 (17), pp. 3916.
  9. Tran T.-S., Tran T.-D., Tran T.-H., et al. (2020), “Synthesis, In Silico and In Vitro Evaluation of Some Flavone Derivatives for Acetylcholinesterase and BACE-1 Inhibitory Activity”, Molecules, 25 (18), pp. 4064.
  10. In this study, no relationship between the structure and the inhibitory effects on BACE-1 of the synthesized derivatives has been drawed, as well as the influence of different substituents at different positions in the structure to the biological activity of the derivatives has not been indicated. These could be done through molecular docking studies. Therefore, to make the research results more convincing, the authors should conduct an additional study of molecular docking to investigate the binding abilities of the compounds to the target (BACE- 1 in this case), which can further explain the biological activities of the derivatives.

From the comments above on the manuscript, the reviewer recommend a major revision for this manuscript before futher consideration.

Author Response

Dear reviewer 2

My name is Tai Kyoung Kim.

Thank you for careful review and suggestions.

We would like to thank you and the reviewers for your insightful comments, which have greatly helped us to improve the quality of our manuscript. And we are honored to submit ‘Molcules’

I wrote this in a hurry due to a problem related to my job, but I have corrected reviewer's opinion as much as possible. I need your help. We admit that our paper had many problems. However, if there is a possibility of approval after viewing the revised paper, please respond quickly. Just answering that there is a possibility of approval, not about approval, would be of great help to me.

Here are our responses to the reviewer’s comments.

Pages and lines are described based on the “Track changes” function.

Reviewer 2 comments.

  1. The authors should check for English and typographical errors throughout the entire manuscript.
  • We conducted a revision of the document. If there are any missing parts, please comment.

  1. Definition for all abbreviations in the text and that included in tables, figures should be given, such as: Aβ, Cbz, HBA, HBD, LPS, NO,…
  • We have added the content related to the abbreviation you pointed out. In addition to the information presented, definitions of all abbreviations are provided.

  1. The abstract should be rewritten to be more condensed and convincing.
  • We revised the content of the abstract as a whole. We have shortened the content and made it easier to understand. (Page1, line13-27)

  1. In these statements “The pathogenesis of Alzheimer's disease (AD) is still unclear, and presently, there are no approved drugs that can be used for its treatment or to stop its progression” (lines 13-14), and “Further, the pathogenesis of the disease is not fully understood, and there are no approved drugs that can be used for its treatment or to stop its progression” (lines 86-88), the underline sentence is not true, and it could be rewritten as “there is no cure for the disease”.
  • I agree with your suggestions, and the text has been revised to actively reflect the opinions of reviewers. (red highlight)

  1. The statement “and can therefore serve as therapeutic agents for the disease” (line 25) should be rewritten as “and can therefore serve as potential therapeutic agents for the disease”.
  • Edited as suggested by the reviewer. (Page 1, line 26-27)

  1. All tables and figures must be placed after their first citations in the text.
  • We corrected th potion of Figure 1 and Table 2 and checked.

  1. In the Table 1, an additional column of molecular weight should be included.
  • Thanks for the reviewer’s comments. MW has been added to Table 1 in consideration of the content.

  1. In the Table 2, the information of a control substance and its BACE-1 inhibitory activity should be included.
  • We used LY2811376 as a stand positive control. Related information has been added to Table 1 and Page 5 line 155.

  1. In Figure 2: “% of contorl” should be corrected into “% of control”. The authors should specify what do the symbols #, **, and * denote for.
  • A typo in Figure 2 has been corrected. In Figure 2, information about the symbols has been added to the lower right coner.

  1. Section 4.2.2: the authors should add the additional information on IR spectra, melting points, and solubilities of synthesized compounds.
  • Added analysis information for IR spectrum in 4.2.2 section. However, we could not enter the information of melting points and solubilities of the compounds. There is no analysis equipment in our laboratory, and analysis within the deadline is not possible. Please understanding.

  1. For fluorine-containing derivatives (RA-2F, RA-4F, RA-24F, RA-PF), coupling constants: JH-Fand JC-F should be added.
    • Coupling constants were calculated and added. However, the JHF value is not clear due to severe overlapping, so it is indicated as multiplet. This related information was added to section 4.2.2, and fluorine-containing derivatives NMR assignment information was added to Supporting information.

  1. IR and MS spectra of synthesized compounds should be included in the Supplementary.
    • Added MS and IR analysis spectrum of all derivatives to the Supporting information.

  1. Scheme 1: The information of the synthetic yields of synthesized compounds should be added.
  • Instead of Scheme 1, the synthesis yield of derivatives is indicated in Figure 1. Please provide your comments in this regard.

  1. The authors should specify which synthesized compound(s) is (are) the new structure(s) with the appropriate citation (ie. Scifinder).
  • The novelty of our proposed compounds were confirmed through Scifinder. It was confirmed that a substance without novelty was included in the derivative. Therefore, the contents of the Discussion have been modified. (page 7, line 187)

  1. Section 4.5. It should be indicated what the final concentrations of the tested compounds were in each well, what control compound was used and its concentrations in the wells.
  • The information of the dirivatives final concentration was added to Section 4.5, line 376

  1. In the discussion, the authors stated that, except for the derivative RA-24F, ramalin and the rest of the derivatives inhibit BACE-1 in a concentration-dependent manner, but the reviewer found no evidence for this statement in the manuscript. Therefore, the authors should provide additional evidence to support this issue.
  • Thanks for the reviewer's comments. However, we would like to solve this problem by deleting the word concentration-dependent in the documents. We want to match the mention of active with “IC50 value at micromolar concentration”, which is in Section 2.3, page 5, line161. Therefore, the contents of the Discussion have been modified (page 7, line196).

  1. The authors should also provide the evidence showing that ramalin derivatives are more stable than ramalin itself.
  • The change in purity of Ramalin and its derivatives was determined by HPLC analysis. Added related results to page 4, line 113.

  1. In the discussion, the authors reported that the tendency to search for small molecules that act simultaneously on multiple therapeutic targets of AD is consistent with the multifactorial nature of AD. Currently, many studies have been conducted for different derivatives such as flavonoids, curcumin and on many targets such as AChE, BACE-1, antioxidant, GSK-3beta, etc. The authors could cite more recent works to support their arguments, such as:

Tran T.-S., Le M.-T., Tran T.-D., Tran T.-H., Thai K.-M. (2020), “Design of Curcumin and Flavonoid Derivatives with Acetylcholinesterase and Beta-Secretase Inhibitory Activities Using in Silico Approaches”, Molecules, 25 (16), pp. 3644.

Tran T.-S., Le M.-T., Nguyen T.-C.-V., et al. (2020), “Synthesis, In Silico and In Vitro Evaluation for Acetylcholinesterase and BACE-1 Inhibitory Activity of Some N-Substituted-4-Phenothiazine-Chalcones”, Molecules, 25 (17), pp. 3916.

Tran T.-S., Tran T.-D., Tran T.-H., et al. (2020), “Synthesis, In Silico and In Vitro Evaluation of Some Flavone Derivatives for Acetylcholinesterase and BACE-1 Inhibitory Activity”, Molecules, 25 (18), pp. 4064.

  • I agree with the reviewer's opinion and have checked the related literature. Thanks for the suggestion. Related papers have been added to Reference.

In this study, no relationship between the structure and the inhibitory effects on BACE-1 of the synthesized derivatives has been drawed, as well as the influence of different substituents at different positions in the structure to the biological activity of the derivatives has not been indicated. These could be done through molecular docking studies. Therefore, to make the research results more convincing, the authors should conduct an additional study of molecular docking to investigate the binding abilities of the compounds to the target (BACE- 1 in this case), which can further explain the biological activities of the derivatives.

  • We plan to continue synthesizing additional Ramalin derivatives. If we identify a structure with good activity among the derivatives, we will do the research suggested by the reviewer. Thanks for your comments.

We revised the paper by actively collecting the opinions of reviewers. Would you please review the revised content and advise if there are any problems.

We look forward to hearing from you.

Best regards,

Dr. Tai Kyoung Kim

Researcher, KOPRI

E-mail: tkkim@kopri.re.kr

Round 2

Reviewer 2 Report

Dear authors,

Thank you very much for revising the manuscript according to the reviewer’s comments. Through the revised manuscript, the reviewer found that the authors made almost all of the revisions according to the reviewer's suggestions, which reflected a great effort by the authors to improve your manuscript. There are still a very few issues that remain unresolved, such as the addition of melting points and solubilities of the synthesized derivatives. However, it is not very important and the reviewer understand your condition. The reviewer also agree with the authors that the relationship between structure and biological effects will be drawn when the authors perform further synthesis of other derivatives and will report in another work.
There are still some minor typographical errors (as shown in Figure 2, upper panel: “contorl”,ect..), which the authors should re-check throughout the entire manuscript.

The reviewer recommend this manuscript to be accepted for publication after correcting minor typographical and spelling errors.

Author Response

Dear Reviewer  2 

Thank you very much for your careful review and prompt reply.

Your review has been very helpful to the content of our paper. 

Thank you for understanding my situation and for accepting the revision I was lacking. 

Best regards,

Dr. Tai Kyoung Kim

Researcher, KOPRI

E-mail: tkkim@kopri.re.kr
